# Additive Manufacturing of Wood Composite Panels for Individual Layer Fabrication (ILF)

**DOI:** 10.3390/polym13193423

**Published:** 2021-10-06

**Authors:** Birger Buschmann, Klaudius Henke, Daniel Talke, Bettina Saile, Carsten Asshoff, Frauke Bunzel

**Affiliations:** 1Chair of Timber Structures and Building Construction, TUM School of Engineering and Design, Technical University of Munich, Arcisstraße 21, 80333 Munich, Germany; henke@tum.de (K.H.); talke@tum.de (D.T.); bettina.saile@tum.de (B.S.); 2Fraunhofer Institute for Wood Research, Wilhelm-Klauditz-Institut WKI, Bienroder Weg 54E, 38108 Brunswick, Germany; carsten.asshoff@wki.fraunhofer.de (C.A.); frauke.bunzel@wki.fraunhofer.de (F.B.)

**Keywords:** binder jetting, sheet lamination, 3D printing, additive manufacturing, adhesive content, pMDI, wood composite, wood particles

## Abstract

The renewable resource, wood, is becoming increasingly popular as a feedstock material for additive manufacturing (AM). It can help make those processes more affordable and reduce their environmental impact. Individual layer fabrication (ILF) is a novel AM process conceived for structural applications. In ILF, parts are formed by laminating thin, individually contoured panels of wood composites which are fabricated additively by binder jetting. The individual fabrication of single panels allows the application of mechanical pressure in manufacturing those board-like elements, leading to a reduction of binder contend and an increase of mechanical strength. In this paper, the ILF process is described in detail, geometric and processing limitations are identified, and the mechanical properties of the intermediate product (panels) are presented. It is shown that the thickness of panels significantly influences the geometric accuracy. Wood composite panels from spruce chips and pMDI adhesive showed flexural strengths between 24.00 and 52.45 MPa with adhesive contents between 6.98 and 17.00 wt %. Thus, the panels meet the mechanical requirements for usage in the European construction industry. Additionally, they have significantly lower binder contents than previously investigated additively manufactured wood composites.

## 1. Introduction

Additive manufacturing (AM) by the use of wood is a way to employ a renewable raw material, virgin or even recycled, which may lead to a reduction of the environmental impact of the manufacturing process. In addition, it can help to achieve specific material properties and reduce material costs. AM by the use of wood has already been investigated in a number of projects [1,2,3,4,5,6]. Potential applications go far beyond prototyping and include furniture production [7], mold making [8], architectural structures [2] as well as biomedical applications such as implants and tissue engineering scaffolds [9].

Binder jetting was among the first processes to be considered as suitable for wood AM [10,11] and is still today a subject of research and development [12,13,14]. A large number of investigations have been reported on wood in fused deposition modeling / fused filament fabrication [15,16,17,18,19,20,21]. However, also granular-based fused deposition processes show high potential for fabricating wood composite parts, especially when the large scale is to be achieved [8,22,23]. Next to these material extrusion processes with thermal reaction bonding, there are some processes that use the extrusion of paste-like mixtures of wood particles and binders [24,25,26]. Furthermore, research has also been executed on the powder bed fusion of wood plastic composites [27]. Using wood in sheet lamination, even though the possibility has already been addressed in a patent on laminated object manufacturing filed in 1995 [28], has not yet found extensive attention [29].

Individual layer fabrication (ILF), a process first described in [30] and the focus of this paper, is not limited to a specific type of material. However, the technique was designed especially with the use of wood in mind. The aim is to obtain additively manufactured wood composite parts with material properties comparable to those of conventionally fabricated wood particle composite boards, as they are widely used in joinery and construction. This goal is to be achieved by introducing a work step of mechanical pressing into the AM process chain. In practice, this is realized by a process in which parts are built up by laminating panels of solid materials. The individual panels are manufactured separately by binder jetting. This allows the application of pressure to these board-like components exactly as in the fabrication of conventional wood composite boards. Depending on the type of binder, either hot or cold pressing can be used. Through the added mechanical pressure, the required amount of the binder can be significantly reduced, and the material strength is increased [31]. The pressing of individual layers results in an identical pressure and temperature for each area of the part, independent of its location inside the building space. This way, homogeneous components with overhangs, bridges, and even unfilled closed cavities can be produced. In addition, manufacturing each layer individually eases the removal of the unbound material, which is an essential issue, especially with particles prone to agglomeration, as in the case of those from wood. 

The fabrication in individual layers seems to bring many advantages, which certainly still has to be proven in detail. The main objective of the work presented in this paper was to quantify the strengths and binder contents achievable by the ILF process at the level of the individual panels. Additionally, addressing the geometric accuracy of the process, the effects of the panel thickness on the contour accuracy of the panels were investigated. Finally, to illustrate the geometric freedom of the process, demonstration objects were fabricated by laminating a series of individual panels.

## 2. Materials and Methods

### 2.1. ILF Process

The ILF process can be described as a combination of the two AM processes, i.e., binder jetting and sheet lamination. As in sheet lamination, parts are formed by laminating thin, individually contoured panels of solid materials. However, in ILF, these panels are not fabricated subtractively, but additively by binder jetting. In addition, after the binder application and before lamination, the sheets are exposed to a mechanical pressure.

Accordingly, the process consists of the following work steps (see Figure 1): Particles are spread in a thin layer on a base by a scattering device (Figure 1a). On this particle bed, a liquid adhesive is applied locally, limited to those areas that are intended to be bound (Figure 1b). The particle adhesive layer is then pressed, potentially under the influence of heat, and a panel with bound and unbound bulk material is obtained (Figure 1c). Finally, the unbound bulk is removed, and the completed panel is transferred and laminated onto the stack of the previously fabricated panels (Figure 1d). Panel production and lamination are repeated, until the desired object is completed. As in all AM processes, the physical fabrication process is preceded by a digital process in which the shape of each single panel is generated, typically by slicing a three-dimensional CAD model. 

It should be noted that the described process chain represents the ILF process in its basic version. Deviating from this the order of the work steps may be altered, the process may be augmented by further work steps, or single-step sequences may be repeated several times before advancing to the next step. For example, the step sequence of bulk scattering, liquid application, and pressing may be repeated several times before advancing to stacking and laminating, so that one layer of the sheet lamination process consists of several layers of the binder jetting process. Alternatively, as another example, a subtractive work step may be introduced after printing and pressing to enhance the contour accuracy of the panels. Finally, as a third example, the unbound bulk material may not be removed after pressing, but only after lamination, acting as a support material for the solid part.

For the investigations presented in this paper, the fabrication of the panels was executed in the following manner: The layer of the scattered wood particles was pressed prior to applying an adhesive in order to prevent particles from being displaced by an adhesive jet. Then, an adhesive was applied locally limited to those areas that were intended to be bound. After adhesive application, a second layer of particles was scattered onto the previous ones to hinder the adhesive from sticking to the press plates. Afterwards, the sandwich was pressed. The steps of liquid application, bulk scattering, and pressing were then repeated, until the desired number of layers or thickness for one panel was reached. Multiple panels were produced this way, stacked by hand and then laminated with an epoxy resin. This resulted in the objects displayed in Figure 2. 

### 2.2. Scattering of Wood Particles

The creation of uniform particle layers of defined height was realized by the use of a specially developed scattering device (see Figure 3b). Figure 3a shows a schematic of the main machine components. Wood particles which are stored in a material hopper, are picked up by the needles of a scatter roller and transported clockwise 90 degrees out of the hopper to a brush roller. An adjustable doctor blade ensures a precise dosing of the material. By rotating counter-clockwise and at a greater speed than that of the scatter roller, the brush roller ensures a break-up of possible agglomerates and an even distribution of particles onto a conveyor belt. The scatter and brush rollers as well as the doctor blade were acquired from IPCO Germany GmbH (Fellbach, Germany). The device can be modified to accommodate a wide range of particle sizes and geometries and allows the precise control of all moving components. By changing the rotational speed of the needle rollers or the linear speed of the conveyor belt, the resulting height of the layer can be adjusted. The scattering width is approximately 500 mm. However, due to a noticeable density drop near the edges, the usable scatter width with an even and consistent particle density is limited to 400 mm.

### 2.3. Dispensing of the Adhesive

In the scope of the overall project, various valve systems for the dispensing of adhesive were tested. The one presented here is an electro-pneumatically driven jet valve system from perfecdos GmbH (Oberhaching, Germany). In this system, a plunger is actuated by a high-frequency, electropneumatic valve. With each down movement, the plunger pushes adhesive through a nozzle and thus produces a droplet of the adhesive with a precisely defined mass. As displayed in Figure 4b, the jet valve was mounted onto a three-axis CNC-portal. This portal moves the valve in parallel tracks over the particle bed. If the frequency of the valve is set high enough, the individual droplets merge into a dispensed line of the adhesive while moving along one track. According to a slice of the CAD data, the valve dispenses adhesive where the final part is to be realized. The resulting pattern is illustrated in Figure 4a.

Multiple parameters can be adjusted in the dispensing system. They can be classified into two groups: mechanically and digitally adjustable parameters. The mechanically adjustable parameters are the nozzle orifice diameter, the plunger thickness, and the plunger retraction height. The plunger thickness and the nozzle orifice can be adjusted by exchanging the individual parts, while the retraction height can be adjusted through a screw. The effect of any mechanical adjustment is variation in droplet mass ($m_{D}$).

Figure 4a shows the digitally adjustable parameters of the dispensing system. These parameters can be varied by setting the respective values in the control program. The plunger movement is defined through the cycle interval (CI). A cycle is a complete downward and upward movement of the plunger. In the time of a cycle interval, the time necessary for the movement itself and the breaks in between the movement are included. The speed the valve moves along the parallel tracks is defined as dispensing speed (v). A greater speed results in a smaller number of droplets dispensed per dispensing line. The distance of the dispensing lines to one another or the distance between two parallel tracks is defined as hatch distance (HD).

Combining the digitally adjustable parameters with the mass of a droplet makes it possible to calculate the theoretical area density ($\rho_{A,a}$) of the adhesive in a panel, as displayed in Equation (1):(1)$$\rho_{A,a}=\frac{m_{D}}{HD*v*CI}$$

With the inclusion of the scattered particles, the theoretical overall density of the adhesive in a panel can be calculated. This can be performed with the use of Equation (2): (2)$$\rho_{a}=\frac{m_{D}}{HD*v*CI*LH}$$

In this context, the layer height (LH) is the overall thickness of a finished panel divided by the number of times adhesive was dispensed.

### 2.4. Materials

The wood particles used for the investigations were spruce wood chips produced by Fraunhofer WKI (Brunswick, Germany). Sieve retention with a mesh size interval between 1.25 and 0.6 mm was used to fractionate the particles. After retention, the mean particle size was measured to be 3.09 mm long and 0.33 mm wide. Wood particles were stored for multiple weeks at the room climate (temperature: 20–25 °C; humidity: 40%–60%) before processing, and the moisture content was identified to be in the range of 7 to 8 wt %.

As an adhesive, the polymeric methylene diphenyl diisocyanate resin pMDI I-BOND^®^ PB EM 4352 (Huntsman Corporation, Salt Lake City, UT, USA) was used. In preliminary experiments, the pMDI adhesive proved to be easily processable, with a viscosity of 220 mPa·s at 25 °C [32], and to have a good wetting behavior, with a surface tension of 41–46 mN/m [33]. Additionally, the cured resin is free of formaldehyde emissions [33]. For the better visibility of the adhesive, a melamine-urea-based pigment of green color with a mean particle size of 3–7 µm was added to the adhesive in a mass ratio of 1:20.

### 2.5. Analytics 

To assess the geometric accuracy of the multi-layer sheet production, three kinds of rectangular cuboids were created by repeating the process steps of scattering, dispensing and pressing multiple times on top of each other. The cuboids each had a base dimension of 15 × 15 mm^2^ and heights of 7, 14, and 21 mm (see Section 3.1). The production parameters are summarized in Table 1. A coloring agent was mixed with the adhesive to enhance its visibility. After printing, the unbound material was removed by means of a steel brush. The dimensions of these objects were identified by three-dimensionally structured light scanning performed with a DAVID SLS-2 3D scanner from Hewlett-Packard (Paolo Alto, CA, USA) and the software Netfabb 2021 from Autodesk. 

To identify the influence of the adhesive dispensing system on the panel’s mechanical properties, an experimental setup was designed by varying the two digitally adjustable parameters, i.e., hatch distance (HD) and dispensing speed (v). They were varied in such a way that, according to Equation (1), theoretical dispensed adhesive variations of 0%, 50%, 100%, and 150% were achieved. Additionally, the orientation of the test specimen in relation to the dispensed lines was set as the third factor. Here, the factor variation consisted of the parallel and perpendicular orientations. The production parameters of the experimental design are summarized in Table 2.

The characterization of the panels mechanical properties was accomplished by flexural and tensile testing. Both the modulus of rupture (MOR) and the modulus of elasticity (MOE) were investigated. Bending tests were conducted according to DIN EN 310 [34] on a Zwick-Roell Z100 (Zwick GmbH & Co. KG, Ulm, Germany), while tensile testing was performed according to DIN EN ISO 527-4 [35] on a Zwick-Roell Z020 (Zwick GmbH & Co. KG, Ulm, Germany). For every parameter set, 4 panels were produced, each with a thickness of 4 mm. Out of every panel, two bending and two tensile specimens were cut out with a mill according to the geometries defined in the respective standards (150 × 50 mm² for bending and type 1B for tensile testing), resulting in a sample size of 8. Adhesive mass fraction was determined by weighing each produced panel and the used adhesive. After cutting, the specimens were conditioned at 20 °C and a 65% relative air humidity to mass constancy for testing. The density of the specimens was determined according to DIN EN 323 [36].

## 3. Results and Discussion

### 3.1. Geometric Analysis 

A significant distortion, which is summarized in Table 3 and displayed in Figure 5a–f, of the planned to real dimensions were observable in all three cuboids. However, the scale of the distortion varied. In the specimen with a height of 7 mm, the maximum width deviated by a factor of 1.65, the maximum horizontal area deviated by a factor of 2.73, and the overall volume deviated from the planned geometry by a factor of 2.0. An increasingly severe deformation was observed in the 14 mm and 21 mm high specimens. Here, the maximum width deviated by a factor of 2.13 and 2.45, the maximum horizontal area deviated by a factor of 4.52 and 5.60, and the overall volume deviated from the planned geometry by a factor of 2.9 and 3.52. Thus, the deformation seemed to increase with the height of the specimen or, to be more precise, the thickness of a panel. In addition, the deformation was not equally distributed along the height, but rather in a curve, with the greatest deviation in the middle and a smaller deviation at the top and bottom of the specimens.

A possible explanation of this deviation can be given through the adhesive spread. The adhesive-dispensing movement was performed according to the planned geometry without any consideration of horizontal adhesive spread caused by intrusion and pressing. However, the curved deviation along the specimen’s height and the increase of deviation with increasing overall height rather seemed to be caused by jammed unbound material. Cross sections in Figure 5g–i shows that the green colored adhesive was primarily located in the 15 mm range it was originally dispensed at. Hence, the deviation was mostly made up of unbound wood particles and not caused by spread of the dispensed adhesive. These unbound particles were most likely jammed together by high pressure, their morphology, and adhesive chemicals intrinsic to the wood [37,38]. Partially bound wood particles could clamp unbound wood particles to the side of the cuboids, which in turn clamped other unbound particles themselves. Thus, a comparatively tough cover was formed. Since higher cuboids had a larger area to clamp unbound particles, the geometric distortion increased with height.

An additional negative effect of increased height can be seen in Figure 5i. The compression of wood particles during pressing was followed by a relaxation or vertical expansion after pressing, which was also influenced by the adhesive [39]. It was observed that the rate of relaxation was greater with unbound than with bound particles. A small height difference was created between the areas with and the areas without adhesive. The result is a lifting force between the bound and unbound areas. This height difference and the resulting force were aggravated through consecutive scattering and dispensing on the same spot. If the lifting force surpassed the interlayer bonding, delamination occurred as observed in the 21 mm specimen. In this spot, the layers were only connected through the unbound, jammed particles.

### 3.2. Mechanical Analysis 

The production of panels with parameter set S4 (see Table 4) was challenging, since the high amount of the adhesive led to the warping of panels and the extensive adhesive spread. Sets S1 to S3 could be produced without any complications. No out-of-the-ordinary spread was observed, and the panels were always of the planned geometry. However, with S4, the resulting panel was noticeably larger and partially deformed. Possibly, an adhesive saturation level was reached between sets S3 and S4, where all cavities between the wood particles were filled with the adhesive. Thus, the excess adhesive in S4 spread sideways under the pressure and deformed the geometry.

The overall amount of the adhesive dispensed from parameter sets S1 to S3 happened according to the prediction with Equation (1). As displayed in Table 4, from both the transition of sets S1 to S2 and S2 to S3, increased amounts of 6.88 and 6.93 grams of the adhesive were dispensed, respectively. This corresponded to an approximate 50% and 100% increase compared to set S1. However, the transition from S3 to S4, with an increase of 8.63 grams of the adhesive, was greater than the previous ones. Thus, the 150% increase of the dispensed adhesive in reference to S1 was slightly exceeded in S4. A possible explanation of this deviation lies in production irregularities or measurement inaccuracy. This is indicated by a comparatively high standard deviation in set S4. Interestingly, the deviation from prediction in set S4 was not as distinct in the adhesive mass fraction. Here, the transitions of all sets were closer to the predicted increase with Equation (1). The density of the produced panels followed the same scheme as the total adhesive mass. While the transitions of S1 to S2 and S3 were identical, S4 had a distinctive higher transition in density. Since the adhesive filled the cavities between the pores inside of the wood particles [40], the panel density was closely linked with the mass of adhesive in the panels. All further mechanical investigations of the panels must therefore not only be viewed as a result of the adhesive mass fraction variation, but also as the variation of panel density.

The results of flexural testing are displayed in Figure 6 as median values with standard deviation as error bars. For the flexural strength, linear increases of strength from sets S1 to S3 were observable in both parallelly (25.95, 32.23, and 39.95 MPa, respectively) and perpendicularly (24.00, 30.75, and 37.00 MPa, respectively) oriented specimens. An even greater increase to 52.45 MPa in the parallelly and that of 51.00 MPa in the perpendicularly oriented specimens can be seen in set S4. Since the adhesive mass fraction and the density of the panels followed the same pattern, a direct proportionality between the flexural strength and the panel density/adhesive mass fraction can be assumed. Similar to the flexural strength, an increase in flexural stiffness can be observed. Here, the values of the perpendicularly oriented S1 specimens increased in the same pattern, from 2.70 GPa (S1), over 3.40 GPa (S2), 4.04 GPa (S3) to 5.30 GPa (S4). The transition from the parallelly oriented S1 to S2 specimens differed slightly from this pattern. Starting at a comparatively high value of 3.04 GPa at S1, the increase to 3.29 GPa in S2 was slightly less pronounced than with the perpendicularly oriented specimens. After this, rises to 3.85 GPa in S3 and 5.34 GPa in S4 were observable. The parallelly oriented specimens seemed to exhibit a slightly higher flexural strength than the perpendicularly oriented ones (maximum: 2.95 MPa for S3). For stiffness, the higher values alternated between parallel and perpendicular orientation (maximum: 0.35 GPa for S1). However, when considering the significant standard deviations of all values (±5.79 MPa strength in parallel S3 and ±0.44 GPa stiffness in perpendicular S1), the actual influence of orientation was debatable.

The increase of mechanical properties with rising density and adhesive content has been investigated in depth in the past [41,42]. Generally, wood adhesive composites fail at the bond line, especially when using wood particles with a large length-to-width ratio [43]. Presumably, if the adhesive content is increased, this bond line is extended and thus strengthened, which in turn strengthens the whole composite. The density of wood adhesive composites is one of the primary adjustment factors for setting mechanical properties of conventional particle boards [31,43]. A higher density leads to fewer voids in the material and thus also fewer elements that could compromise mechanical properties.

In Figure 6, next to the flexural properties of the panels, the required flexural strength and stiffness (lower 5% fracile limit) of particleboards according to DIN EN 312 [44] are displayed. The lowest requirement of the flexural strength for P1 (general purpose in the dry climate) class thin boards (4–6 mm) lied at 11.50 MPa and at a 1.80 GPa flexural stiffness for P2 (interior decoration and furniture) class boards. The requirements for highly durable P7 (severe stresses and high load-bearing capacity in the wet climate) class boards were at a 21 MPa flexural strength and a 3.10 GPa stiffness. Considering a normal distributed variance of the panel’s flexural properties [43], for a sample size of 8, the lower 5% fracile limit is equal to 1.89 times the standard deviation [45]. Thus, panels produced with parameter set S1, with a 5% fractile value of 16.76 MPa, fulfilled the requirements of P4 (load bearing purposes in the dry climate) class particle boards regarding the flexural strength. Panels from set S2 onward fulfilled the flexural strength requirements of highly durable P7 class boards. Concerning the flexural stiffness, S1 panels only fulfilled the requirements of P1 class boards with a 5% fractile value of 1.87 GPa. S2 panels can be classified as P5 (load bearing purposes in the damp climate) class with 2.57 GPa and from S3 onward, regarding mechanical properties, a classification as P7 is possible.

In comparison to other additively manufactured wood products, ILF panels showed a relatively high flexural strength. Ayrilmis et al. [18] observed that dense wood-PLA objects created by fused filament fabrication (FFF) and a wood content of 30–40 wt % had an ultimate flexural strength of 33.2 MPa. Parts produced via paste deposition had, with concrete as a binder and a wood content of 13.6 wt %, a flexural strength of 4.08 MPa [24], whereas with urea-formaldehyde as a binder and a wood content of 15 wt %, a strength value of 19.00 MPa was reached [25]. For paste extrusion, as well as FFF, an increasing wood content significantly increases the necessary force for extrusion [21,25]. This makes the processing of high wood contents difficult for these AM processes.

The tensile properties of panels, displayed in Figure 7 with median values and standard deviations as error bars, showed a similar behavior as flexural properties when increasing the panel density and the adhesive mass fraction. A linear increase of tensile strength in the perpendicularly oriented specimens was observed. They exhibited tensile strengths of 17.50 MPa when produced with parameter set S1, 20.85 MPa with S2, 27.05 MPa with S3, and 33.40 MPa with S4. The parallelly oriented specimens showed a less linear increase of tensile strength. The specimens produced with set S1 displayed strengths of 16.35 MPa, 22.05 MPa with S2, 24.45 MPa with S3, and 34.45 MPa with S4. Seemingly, an increase of the adhesive in the dispensing lines (S1–S2 and S3–S4) led to a higher rate of increase in tensile strength than an overall increase of adhesive by decreasing the hatch distance (S1–S3 and S2–S4). The same observation can be made for the tensile stiffness. Here, the perpendicularly oriented specimens exhibited values of 3.76 GPa when produced with S1, 4.03 GPa with S2, 4.85 GPa with S3, and 5.55 GPa with S4. The parallelly oriented ones had values of 3.49 GPa with S1, 4.12 GPa with S2, 4.49 GPa with S3, and 5.38 GPa with S4. As with the flexural properties, this orientation-dependent variation of tensile properties was debatable, when comparing it with a high standard deviation. However, it should be kept in mind when planning with the dispensing pattern according to Equation (1) detailed in Section 2.4.

The classification scheme of DIN EN 312 [44] does not include tensile properties as qualification criteria. Thus, the characteristic values of tensile properties of particle boards [31,43] were included in Figure 7 as a reference. The tensile strength values of conventional particle boards were between 8.00 MPa and 10.00 MPa, and the stiffness is in an approximate range of 2.50 GPa and 3.00 GPa. All values of the panels tensile properties for all parameter sets and orientations were greater than these characteristic values.

Objects produced by FFF with a low wood content have higher tensile strength values than ILF panels. Karitz et al. [21] showed that PLA filament with a wood content of only 10 wt % had an increased tensile strength (57 MPa) in comparison to pure PLA (55 MPa). However, at a 50 wt % content of wood fibers, the tensile strength dropped to 30 MPa. This value was surpassed by the ILF panels with a wood particle content of 83 wt %. Thermoplastic products, such as those made with FFF or powder bed fusion, rely on a polymer matrix to achieve high mechanical properties. Any larger addition of wood particles weakens this matrix [15]. It is presumed that with ILF the intrinsic properties of wood can be utilized to a higher degree. Here, the mechanical properties are rather a result of the wood particles’ strength and are compromised by voids and insufficient bonding between them [43].

## 4. Conclusions

A new process of additively producing wood composite parts with a low adhesive content and high mechanical properties was presented. The relevant machinery and processing strategies were detailed and explained. Additionally, first test objects were created, and their geometric and mechanical properties were investigated. The geometric accuracy of panels was discovered to be significantly dependent on the panels’ thickness. While thinner panels with a height of 7 mm could be produced effortlessly, thicker panels with a height of 21 mm showed severe geometric distortion and compromised integrity trough delamination. Mechanical properties (flexural and tensile) were analyzed as a function of adhesive content, panel density, and orientation relative to the adhesive-dispensing pattern. The orientation to dispensing pattern was identified to have only a minor effect on flexural and tensile properties. The adhesive content and density of the panels had a significant effect on mechanical properties. A 150 % increase of the adhesive content, in combination with the related increase in density, led to a doubling of flexural and tensile strengths. With an adhesive mass fraction of 17 wt % panels with up to a 52.45 MPa flexural strength and a 34.45 MPa tensile strength could be manufactured. All the produced panels fulfilled the flexural requirements for usage as particle boards in construction industry and largely even surpassed the requirements of the most demanding class. It was also demonstrated that the mechanical properties of the panels could be modified solely by altering the control program. Thus, not only individually contoured, but also individually graded wood composite panels can be fabricated with ILF.

## Figures and Tables

**Figure 1 polymers-13-03423-f001:**
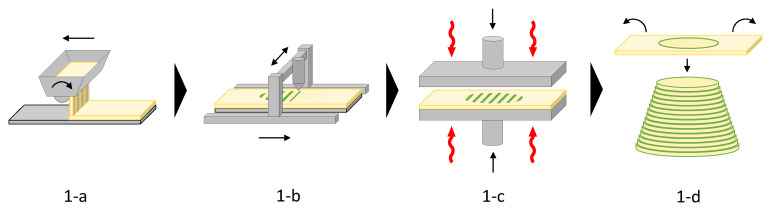
Schematic of the basic individual layer fabrication (ILF) process. Particles are spread in a thin layer on a base by a scattering device (**a**). On this particle bed, a liquid adhesive is applied locally, limited to those areas that are intended to be bound (**b**). The particle adhesive layer is then pressed, potentially under the influence of heat, and a panel with bound and unbound bulk material is obtained (**c**). Finally, the unbound bulk is removed, and the completed panel is transferred and laminated onto the stack of the previously fabricated panels (**d**).

**Figure 2 polymers-13-03423-f002:**
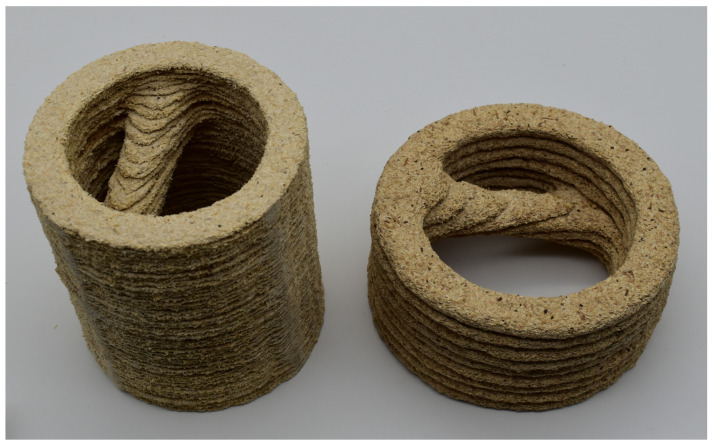
Demonstration objects for the ILF process with different panel thicknesses. While the panels were produced as described in the following sections, stacking and lamination were performed by hand with an epoxy resin as a binder.

**Figure 3 polymers-13-03423-f003:**
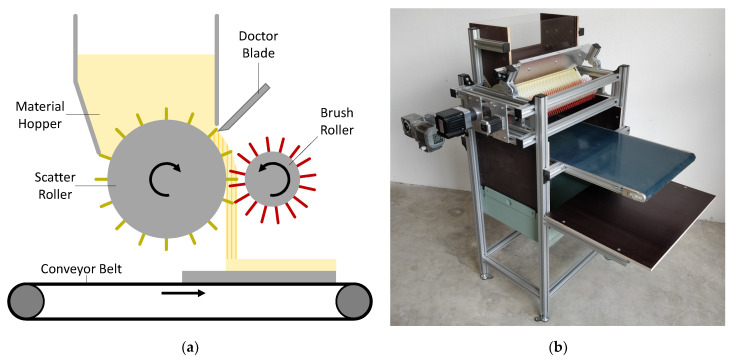
(**a**) Schematic of the scattering device; (**b**) image of the scattering device.

**Figure 4 polymers-13-03423-f004:**
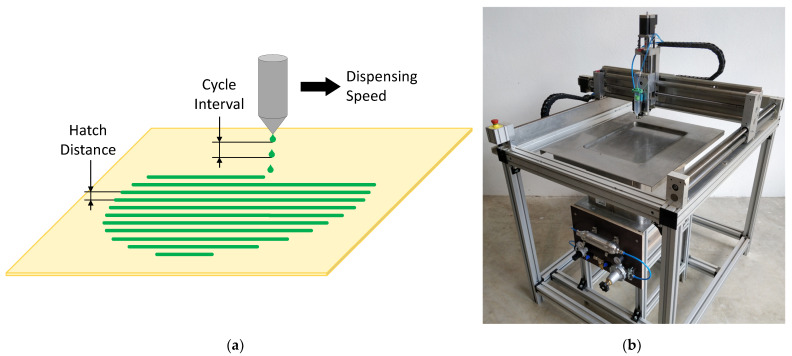
(**a**) Schematic of the dispensing pattern and digitally adjustable process parameters; (**b**) image of the dispensing device.

**Figure 5 polymers-13-03423-f005:**
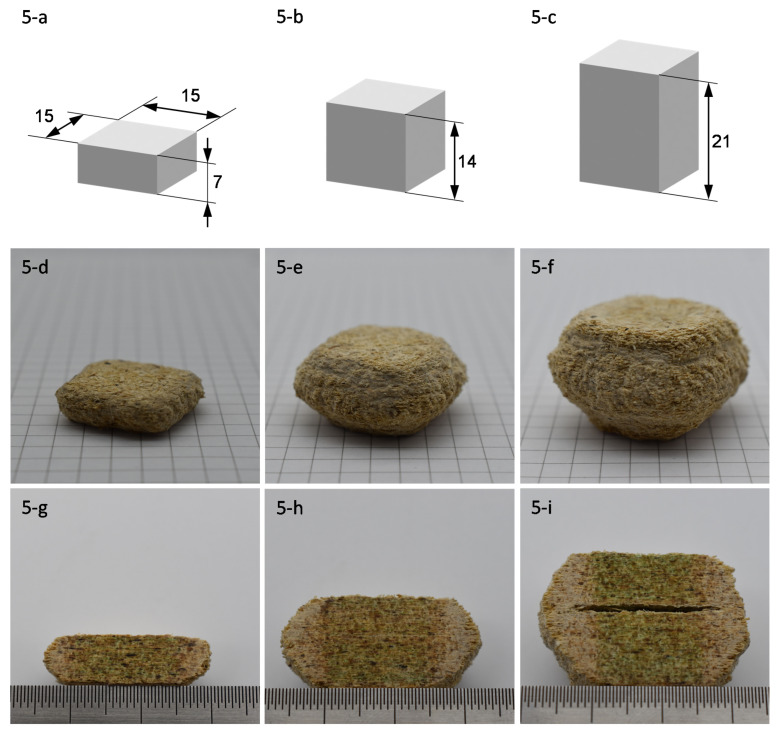
(**a**–**c**): Original CAD data of the planned rectangular cuboid specimens with a height of 7, 14, and 21 mm; (**d**–**f**) produced cuboids (background: standard 5 mm graph paper); (**g**–**i**) cross-sections of the specimens (distance between large ticks: 10 mm).

**Figure 6 polymers-13-03423-f006:**
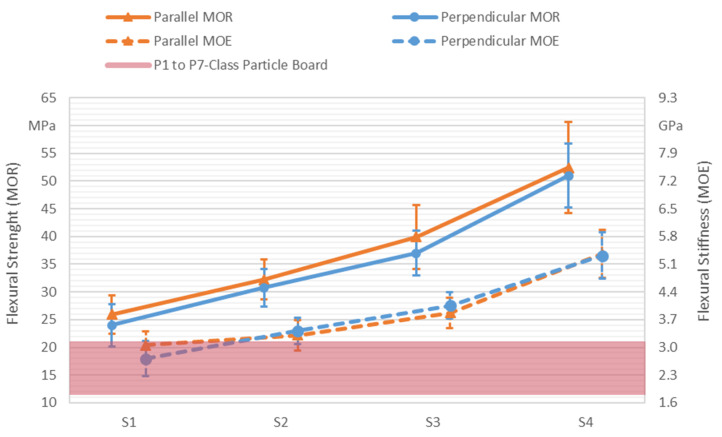
Results of flexural testing. Solid lines display values of flexural strength (modulus of rupture (MOR)) and dashed lines values of flexural stiffness (modulus of elasticity (MOE)). Orange triangles indicate the parallel orientations of specimens, and blue circles represent the perpendicular orientations of specimens. The red area contains requirements for P1 to P7 particle boards.

**Figure 7 polymers-13-03423-f007:**
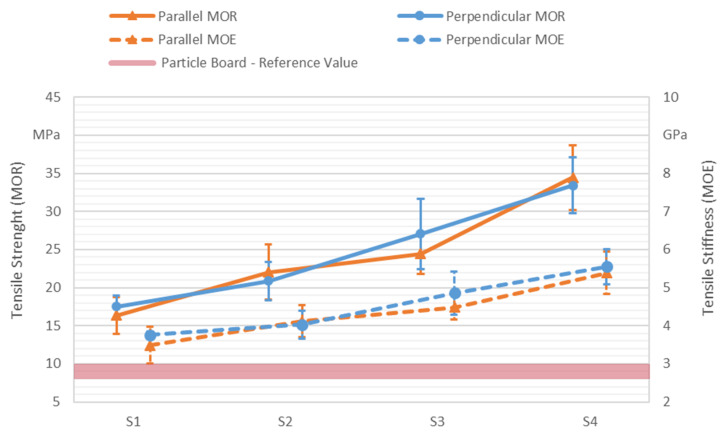
Results of tensile testing. Solid lines display values of the tensile strength (MOR), and dashed lines indicate values of the tensile stiffness (MOE). Orange triangles indicate the parallel orientation of specimens, and blue circles represent the perpendicular orientation of specimens. The red area contains the reference values of classical particle boards.

**Table 1 polymers-13-03423-t001:** Production parameters of the rectangular cuboids.

	Parameter	Value
Scattering	Layer height	0.35 mm
Dispensing	Plunger thickness	1.5 mm
	Nozzle orifice	150 µm
	Plunger retraction	600 µm
	Hatch distance	2 mm
	Dispensing speed	150 mm/s
	Cycle interval	6 ms
Pressing	Duration	90 s
	Temperature	180 °C
	Pressure	55 bar

**Table 2 polymers-13-03423-t002:** Production parameters for the flexural and tensile specimens. Parameters not specified are identical to the ones presented in Table 1.

	Parameter			Value	Adhesive Variation
Scattering	Layer height			0.5 mm	---
Dispensing	Hatch distance (*HD*)	-	Dispensing speed (*v*)	4 mm; 150 mm/s	+0%
				4 mm; 100 mm/s	+50%
				2 mm; 150 mm/s	+100%
				2 mm; 100 mm/s	+150%
	Orientation			Parallel	---
				Perpendicular	---

**Table 3 polymers-13-03423-t003:** Dimensions of the produced rectangular cuboids and their comparison with those of the planned geometry.

Height of the Specimen	Dimension	Planned	Real (Max.)	Factor of Deviation
7 mm	Width	1.50 cm	2.48 cm	1.65
	Horizontal area	2.25 cm²	6.15 cm²	2.73
	Overall volume	1.58 cm³	3.16 cm³	2.00
14 mm	Width	1.50 cm	3.20 cm	2.13
	Horizontal area	2.25 cm²	10.17 cm²	4.52
	Overall volume	3.15 cm³	9.15 cm³	2.90
21 mm	Width	1.50 cm	3.68 cm	2.45
	Horizontal area	2.25 cm²	12.60 cm²	5.60
	Overall volume	4.73 cm³	16.63 cm³	3.52

**Table 4 polymers-13-03423-t004:** Summary of the dispensed adhesive and the density values of the panels. Values in brackets are the standard deviations (SD).

Parameter Set: *HD*; *v*	Mass of the Dispensed Adhesive in g (SD)	Adhesive Mass Fraction in wt % (SD)	Panel Density in kg/m³ (SD)
S1: 4 mm; 150 mm/s	13.49 (±1.0)	6.98 (±0.5)	808 (±24.6)
S2: 4 mm; 100 mm/s	20.37 (±0.6)	10.28 (±0.3)	850 (±28.7)
S3: 2 mm; 150 mm/s	27.30 (±0.5)	13.32 (±0.4)	890 (±14.9)
S4: 2 mm; 100 mm/s	35.99 (±1.7)	17.00 (±0.6)	958 (±33.9)

## Data Availability

The data presented in this study are available on request from the corresponding author.

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
