# Peer review of "Additive Manufacturing of Wood Composite Panels for Individual Layer Fabrication (ILF)"

_polymers, 2021, doi:10.3390/polym13193423_

Round 1
Reviewer 1 Report
The authors present a paper describing a new additive manufacturing technique of wood composite panels for individual layer fabrication. Overall, the work is interesting and well organized. In my opinion, the paper presents conditions to be considered for publication, needing, however, some revisions that do not compromise the overall quality of the work.
In the introduction, the authors adequately present the framework of the work carried out, however, the part referring to the literature review could be a little more developed, not only referring to the technique, but also the products obtained.
The information regarding mix proportions is very brief. It should be noted that mix proportions also have a great influence on the results presented, in this sense, this part should be more detailed.
In verifying the elements produced, the authors limit themselves to performing a geometric and mechanical verification in relation to a set of allegedly reference values. However, it would be interesting to understand the differences from other techniques. Is this system actually better? It would be interesting to benchmark the results obtained. The mix proportions of the different materials also influence the results. The authors should mention something about this fact.
Author Response
Dear Mrs/Mr Reviewer,
thank you so much for your Input! We will try our best to implement your suggestions into our manuscript.
However we are somewhat usure what you mean with "mix proportions". Are you referring to the adhesive to pigment ratio? A further elaboration of this comment would be greatly appreciated.
with best regards
Birger Buschmann
Reviewer 2 Report
The manuscript is focused on an innovative and interesting research topic, namely development and characterization of wood-based panels (particleboards) produced using additive manufacturing method. In this respect, the topic of the manuscript is relevant and appropriate to the aims and objectives of the Special Issue "3D Printing in Wood Science" of Polymers Journal.
In general, the abstract (lines 11-20) and the keywords (lines 21-22) correspond to the title, aims and objectives of the manuscript.
I’d recommend to the authors to be more specific in the abstract by providing the exact aim of the presented research work, as well as to include some specific results of the study, e.g. related to the mechanical properties of the developed composites.
Overall, the Introduction part is well-written and informative, but can be further elaborated by adding more relevant references. Example: in lines 28-29, the authors stated “A large number of investigations has been reported…”, and then there is only one reference included. Please, refer to the following recent interesting articles in the field of additive manufacturing in wood science:
doi:10.32604/jrm.2021.016128
https://doi.org/10.3390/polym13010144
10.1007/s00170-021-07382-y
10.3390/polym13081211v
https://doi.org/10.1080/1023666X.2019.1651547
https://doi.org/10.3390/ma10040339
In addition, I’d recommend to the authors to provide the specific objective of the research at the end of the Introduction part.
In line 106, please provide some information on the specialized scattering device. If it is a laboratory prototype, please add this in the description.
In line 124, please add information about the jet valve system used (company producer, city, country).
In line 159, please add relevant information about the sieve equipment used for carrying out the sieve analysis.
In line 161, please be more specific about the “room climate” by providing the temperature and relative humidity. Please also provide the period of conditioning.
In line 164, please provide relevant information and characteristics of the pMDI used as a binder. Please explain the selection of pMDI as an adhesive.
In line 171, page 5, the authors refer to Figure 5, which however is located on page 8 (line 241). This makes reading the manuscript a bit difficult. Please put Figure 5 (5a to 5c) in point 2.5. Analytics, where it was referenced.
In lines 192 and 193, please provide information about the universal testing machines used for conducting the mechanical properties tests - Zwick-Roell 192 Z100, and Zwick-Roell 193 Z020, e.g. company producer, city, country).
In line 201, I’d recommend to revise point 3 from “Results” to “Results and Discussion”.
I’d recommend to the authors to add a short description of the referenced types of particleboards (P1 to P7), e.g. P1: general purpose boards for use in dry conditions.
In general, the Results section is well-structured, specific and informative, but more discussion with previously published research works is recommended.
The Conclusions (lines 343-363) are consistent with the results and reflect the main findings of the study.
The references cited are appropriate and correspond to the topic of the manuscript. The inclusion of additional references, especially in the Introduction and Results sections of the manuscript is recommended. This will increase the value of the presented work.
Best regards!
Author Response
Dear Mrs/Mr Reviewer,
thank you so much for your elaborate Input! We have tried our best to implement your suggestions into the manuscript.
with best regards
Birger Buschmann
Round 2
Reviewer 1 Report
The authors amended the paper as per the reviewers' recommendations. In my opinion, the paper presents conditions to be considered for publication.